# A Novel Low Complexity Two-Stage Tone Reservation Scheme for PAPR Reduction in OFDM Systems

**DOI:** 10.3390/s23020950

**Published:** 2023-01-13

**Authors:** Yung-Ping Tu, Chiao-Che Chang

**Affiliations:** Department of Electronic Engineering, National Formosa University, Yunlin 632301, Taiwan

**Keywords:** orthogonal frequency division multiplexing (OFDM), tone reservation (TR), peak-to-average power ratio (PAPR), peak reduction tone (PRT), complementary cumulative distribution function (CCDF)

## Abstract

Orthogonal frequency division multiplexing (OFDM) has the characteristics of high spectrum efficiency and excellent anti-multipath interference ability. It is the most popular and mature technology currently in wireless communication. However, OFDM is a multi-carrier system, which inevitably has the problem of a high peak-to-average power ratio (PAPR), and s signal with too high PAPR is prone to distortion when passing through an amplifier due to nonlinearity. To address the troubles caused by high PAPR, we proposed an improved tone reservation (I-TR) algorithm to alleviate the above native phenomenon, which will pay some modest pre-calculations to estimate the rough proportion of peak reduction tone (PRT) to determine the appropriate output power allocation threshold then utilize a few iterations to converge to the near-optimal PAPR. Furthermore, our proposed scheme significantly outperforms previous works in terms of PAPR performance and computational complexity, such as selective mapping (SLM), partial transmission sequence (PTS), TR, tone injection (TI), etc. The simulation results show that in our proposed scheme, the PAPR is appreciably reduced by about 6.44 dB compared with the original OFDM technique at complementary cumulative distribution function (CCDF) equal to $10^{-3}$, and the complexity of I-TR has reduced by approximately 96% compared to TR. Besides, as for bit error rate (BER), our proposed method always outperforms the original OFDM without any sacrifice.

## 1. Introduction

At present, the Internet of Things (IoT) of 5G [1,2,3] is one of the critical developments of wireless sensing networks [4,5,6]. In addition, the growing popularity of the Internet of Things (IoT) applications includes smart homes, industrial automation, traffic auxiliary management, healthcare, and crisis response for natural and artificial disaster prevention. Namely, efficient power management for wireless sensing transmission services has become an extremely important issue of Internet of Things (IoT) information delivery. If there is no proper power management and control, it will cause nonlinear amplification and distortion of the transmitted signal, especially in a multi-carrier environment [7,8]. Furthermore, due to OFDM technology possessing excellent spectral efficiency and the ability to resist inter-symbol interference caused by multi-path propagation [9,10], it has become one of the most mainstream wireless communication technology. It is a pity, that due to OFDM being a multi-carrier system, it is prone to cause high PAPR. In addition, in case the PAPR of the signal is too high, the signal transmission quality will be significant distortion. By then, a more complicated and expensive amplifier design within the transmitter is required to take precautions against the trauma due to the high PAPR, which is an undesired circumstance. Therefore, searching for an efficient method to reduce PAPR in OFDM systems has become a great urgent research field.

Nowadays, many researchers have proposed various well-known technologies to reduce PAPR in OFDM systems [11,12,13], such as clipping, SLM, PTS, TI, TR, etc. Among them, the clipping technique [14,15] is the easiest to implement, because only the input signal needs to be limited, but the clipped signal is prone to nonlinear distortion, and the location and size of the clipping are difficult to decide and store, which results in BER performance degradation [16] and spectral efficiency decline. In the SLM scheme [17,18,19], the frequency domain signals are multiplied by the rotation factor vector and generate several candidate signals, and the one with the minimum PAPR is selected for transmission.

As for the PTS technique [20,21], the signal is divided into multiple clusters, and an Inverse Fast Fourier Transform (IFFT) is performed on each cluster and multiplied by a rotation factor to change the phase of the signal, and all clusters are summed and combined into the output signal, then the last step is to find the signal output with the optimal PAPR by searching for different rotation factors. Moreover, in ref. [20], Zeid T. Ibraheem et al. discuss the PAPR caused by the different partitioning algorithms in the PTS technique, and the simulation results are given. For the four symbols of QPSK, 8-PSK, 16-QAM, and 64-QAM, four partitions are adopted, such as uniform adjacent, interleaved, pseudo-random, and their proposed non-uniform adjacent algorithm, in which the PAPR performance of pseudo-random partition is slightly better than the other three. Additionally, in ref. [21], Kuo-Chen Chung et al. utilize the codeword of linear code as a rotation factor to reduce the complexity of PTS technology and provide error correction capability. Furthermore, many researchers have struggled to vary the above techniques to optimize PAPR or reduce the computational complexity [21,22,23,24,25,26]. As in ref. [22], to avoid waste calculations repeating, Tanairat Mata et al. replicated the same cluster as other operations and employed the Artificial Bee Colony (ABC) algorithm instead of the original global search method, which they said can reduce computational complexity. Practically, doubling the number of clusters means doubling the number of IFFTs, which invisibly results in an unacceptably large amount of computational complexity. To fundamentally solve the problem of computational complexity, in ref. [23], Ji Ce et al. studied the repetition characteristics of the rotation factor in the interleave partition scheme of the PTS technology, where PTS uses the interleaving partition method to divide four clusters and provides four rotation factors to choose from. Meanwhile, the number of rotation factor combinations can be equivalent to 1/16 of all candidate groups, and the PAPR performance is the same as the searching scheme.

Meanwhile, the TI technology in ref. [27] extends the constellation point locations according to the original constellation, that is, it provides each point of the original constellation with more options and diversity to map to other equivalent points in the constellation diagram. As for the traditional TR method [28] proposed by Sung-Eun Park et al., the controller of the TR method selects a suitable peak cancellation signal according to different lengths and positions and inserts it into the original signal to reduce PAPR. To optimize PAPR performance, a multi-stage TR scheme was proposed by Dae-Woon Lim et al. [29], who selected some suitable signals from the candidate PRT set and performed multiple TR methods. However, when using the above technology it is difficult to reconcile the conflict between the lower PAPR and the lower complexity, among which the TR scheme is relatively feasible. However, it must be mentioned that the optimal PRT set selection for the TR scheme is a non-deterministic polynomial-time (NP) problem that cannot be solved. For example, in ref. [30] Fang, W. et al., according to the degree of PAPR that each tone contributes to the decrease, proposed a scheme to measure the amount of PRT, which is about 1.46% of the subcarriers.

As for our proposed scheme, we first utilize the Monte Carlo method through the analysis of probability and statistics procedure to estimate the PRT proportion of the transmitted signal and then brings forward a reliable and succinct statistical relationship with the subcarrier number, *N*, of the multiple carriers environments. Simultaneously, we produce a rough power allocation threshold, $\beta_{rough}$, which plays an initial role in the iterative procedure, and that moreover can reduce the iteration number in the second stage. Next, the second stage in line with the rough power allocation threshold proceeds a small number of iterations to amend the power allocation threshold and assorts reasonable power to the transmit signal to refine the PAPR.

The contributions of this article are as follows:We use the Monte Carlo method to bring forward a reasonable and laconic amount of PRT estimates, which is 0.08 multiplied by the number of subcarriers, *N*.We estimated a rough power threshold in the initial stage to diminish the PAPR refined iteration number in the second stage, furthermore, it can significantly reduce the total complexity.We utilize an iteration scheme to balance the peak power and average power to obtain a near-optimum PAPR.We apply the side information to the auxiliary receiver to preserve BER performance.

The rest of this paper is organized as follows. Section 2 introduces the system model of this study as well as previous famous techniques. Section 3 illustrates the proposed scheme. Section 4 provides the simulation results and computational complexity analysis to verify and discuss the comparison of our proposed scheme with the other previous works. Section 5 summarizes the study.

## 2. System Model

In this paper, uppercase and lowercase symbols represent frequency-domain signals and time-domain signals, respectively. Besides, the bold font denotes a vector.

### 2.1. OFDM System and PAPR Definition

Figure 1 shows a block diagram of the basic architecture of an OFDM system. Within this, assume that the serial signal modulated by QAM/PSK will be converted into a parallel signal $X=\left[X_{0},X_{1},\ldots ,X_{N-1}\right]$, where *N* is the number of subcarriers, and then X is processed by IFFT to produce a Discrete OFDM signal, x. Therefore, a set of OFDM signals $x=\left[x_{0},x_{1},\ldots ,x_{N-1}\right]$ of length *N*, where $x_{n}$ is the $n^{th}$ tone, can be expressed as:(1)$$x_{n}=\frac{1}{\sqrt{N}}\sum_{k=0}^{N-1}X_{k}e^{j\frac{2\pi kn}{N}};\;0\leq n\leq N-1$$

The modulated OFDM signal is emitted to the Gaussian channel with the additive white Gaussian noise (AWGN) through a high-power amplifier (HPA), and the receiver restores the signal to the original signal through FFT and QAM/PSK demodulator.

Apart from this, the PAPR definition of an OFDM signal can be expressed by the following formula [11]:(2)$$PAPR=\frac{\underset{0\leq n\leq N-1}{max}\left[{\left|x_{n}\right|}^{2}\right]}{E\left[{\left|x\right|}^{2}\right]}$$

The numerator and denominator are the maximum peak power and average power of the output time-domain complex signal, respectively. In this paper, we use the CCDF to evaluate PAPR quality:(3)$$CCDF=P\left(PAPR>\alpha \right)=1-{\left(1-e^{-\alpha }\right)}^{N}$$
where α is a threshold. The CCDF performance expresses the probability of a PAPR beyond a threshold.

### 2.2. Related Work

To promote completeness and readability. In this subsection, we explain the existing methods in more detail. It includes selective mapping (SLM), partial transmit sequence (PTS), tone injection (TI), and tone reservation (TR).

#### 2.2.1. Selective Mapping (SLM)

The selective mapping (SLM) method is one of the schemes used to reduce PAPR in OFDM system, and its block diagram is shown in Figure 2 [18]. In the SLM algorithm, the input signal sequence $X=\left[X_{0},X_{1},\ldots ,X_{N-1}\right]$ and the rotation factor group $P_{v}=\left[P_{v,0},P_{v,1},\ldots ,P_{v,N-1}\right]$, 1≤v≤V by component-wise vector multiplication can obtain $X_{v}$ as follows:(4)$$\begin{array}{cc} X_{v} & =X⊗P_{v}\\ \; & =\left[X_{0}P_{v,0},X_{1}P_{v,1},\ldots ,X_{N-1}P_{v,N-1}\right]\\ \; & =\left[X_{v,0},X_{v,1},\ldots ,X_{v,N-1}\right],\;1\leq v\leq V \end{array}$$
where ⊗ is component-wise multiplication of two vectors, and *V* is the divide group number, and $P_{v,n}=e^{j\phi_{v,n}}$, $\phi_{v,n}\in \left[0,2\pi \right)$. In general, $P_{v,n}$ using binary or quaternary elements, that is, {+1,−1} or {+1,−1,+j,−j}, where $j=\sqrt{-1}$. Next, perform IFFT operation on *V* input symbol sequences $\left\{X_{1},X_{2},\ldots ,X_{V}\right\}$, and select the signal output $\tilde{x}$ with the best PAPR performance. We can denote $\tilde{x}$ as follows:(5)$$\tilde{x}=\underset{1\leq v\leq V}{min}\left\{PAPR\left[IFFT\left(X_{v}\right)\right]\right\}$$

Obviously, as the group number increases, PAPR performance also improves, but a larger *V* also means higher computational complexity. Besides, the selected best rotation factor combination will also be transmitted to the receiver.

#### 2.2.2. Partial Transmission Sequence (PTS)

The concept of the partial transmit sequence (PTS) scheme is to partition the input signal into disjoint clusters, which will be phase adjusted by the rotation factor, afterwards, the amended cluster signal is summed to produce a PTS candidate signal. Finally, the lowest PAPR is selected from the candidate sequence set for transmission.

The block diagram of PTS technology is shown in Figure 3 [21]. First, the input *N* subcarriers are divided into *V* disjoint clusters, as follows:(6)$$X=X_{0}+X_{1}+\cdots +X_{V-1},\;0\leq v\leq V-1$$
where $X_{v}$ can be a cluster of continuous located, interleave located, or pseudo-random located, and the size is equalized through zero padding. Next, each cluster is summed after the IFFT operation and multiplied by the corresponding rotation factor. In other words,
(7)$$\tilde{x}=IFFT\left\{\sum_{v=0}^{V-1}b_{v}X_{v}\right\}=\sum_{v=0}^{V-1}b_{v}\cdot IFFT\left\{X_{v}\right\}=\sum_{v=0}^{V-1}b_{v}x_{v}$$
where $\tilde{x}$ is the PTS output signal matched to the PAPR minimized set of rotation factors. $b=\left[b_{0},b_{1},\ldots ,b_{V-1}\right]=\left\{e^{j2\pi l/W}|l=0,1,\ldots ,W-1\right\}$ is the selected rotation factor, and *W* is the number of equal divisions of the phasor, that is, the number of optional phasor for the rotation factor. Here, the number of rotation factor sets and clusters will directly affect the computational complexity and performance of PAPR, especially as the number of clusters increases and as the computational complexity will increase exponentially.

#### 2.2.3. Tone Injection (TI)

Figure 4 shows a block diagram for the TI technique [27]. The basic concept of the TI technique is to increase the constellation points so that each of the points in the original constellation can be mapped into other equivalent points in the expanded constellation, where altering the constellation point can be utilized for PAPR reduction. Specifically, Figure 5 is an example of a 16-QAM being mapped to a wider constellation 9 × 16-QAM, and the transmitter must select one of the best PAPR performances from A0 to A8 to transport the original data, that is, A0.

Moreover, the TI technique does not need to sacrifice data rate while reducing PAPR, in other words, TI signal $\tilde{x}$ is the combination of the data signal x and PRT signal c. We can present $\tilde{x}$ as follows:(8)$$\begin{array}{cc} \tilde{x} & =x+c\\ \; & =IFFT\{X+C\} \end{array}$$

TI technology does not need to waste extra transmission sequences to reduce PAPR, because the auxiliary signal has been integrated into the transmission signal sequence. Although TI is a seemingly perfect method, the IFFT operation caused by finding the optimal constellation point will increase the computational complexity, and with denser points on the constellation, makes signal recovery more difficult, predictably sacrificing BER performance.

#### 2.2.4. Tone Reservation (TR)

The block diagram of traditional TR technology is shown in Figure 6 [28]. First, we select some bits that are reserved as PRT vector, which is denoted by C. Within that, the controller assigns PRT vectors with different lengths and positions to insert into the data information X, and a frequency-domain TR signal S can be obtained. It is worth mentioning that X and C are nonoverlapped in the frequency domain.

In contrast to the time-domain, the TR signal consists of a PRT vector c and data information x, where c is considered as a peak countervail signal since it is used to offset the excessively high peak value in the time-domain signal to achieve the purpose of reducing PAPR. Therefore, the TR signal in the time domain can be expressed as:(9)$$s_{n}=x_{n}+c_{n}=IFFT\left(X_{n}+C_{n}\right);\;0\leq n\leq N-1$$

Likewise, in the TR scheme, PAPR can be redefined as:(10)$$PAPR_{TR}=\frac{\underset{0\leq n\leq N-1}{max}\left[{\left|s_{n}\right|}^{2}\right]}{E\left[{\left|s\right|}^{2}\right]}$$

## 3. Proposed Scheme

We know that solving the trade-off between reducing PAPR and lowering computational complexity in OFDM systems is an urgent priority. Therefore, we integrate an iterative algorithm into the TR scheme to obtain the near-optimal power allocation and alleviate the embarrassment of the PAPR issue, which is named the I-TR scheme in this paper. To convenient readability and enhance clarity, we list and describe important notices and parameters within the proposed method in Table 1 before detailing them.

In our proposed I-TR scheme, to improve the PAPR performance, we perform power attenuation at the positive peak and power boost at the negative peak according to the PAPR definition, so that the gap between the signal peak and the average value gradually shrinks in the iterative process. To this end, finding a suitable power allocation threshold β for tones is the primary goal, in other words, the higher the credibility of β, the more performance can be improved when the powers are amended using β.

Therefore, to reduce complexity and promote PAPR performance, we divide the I-TR scheme into two stages to complete. The first stage is shown in Figure 7a, where the power sorting vector $x^{\prime }$ is used as the power allocation threshold, and the pre-calculate procedure is sequentially performed, then a rough power allocation threshold $\beta_{rough}$ and the coarse amount of PRTs can be produced, simultaneously. It must be mentioned, that this $\beta_{rough}$ is not yet the optimal power allocation threshold but a sub-optimal one. The second stage is shown in Figure 7b, and we know that using the output $\beta_{rough}$ of the first stage to play a suitable initial role is beneficial to reducing iteration number. Thus, via $\beta_{rough}$ we perform a binary search and iterate to obtain a refined PAPR value until convergence.

For more detail and clear illustrations of the algorithm of the first stage and the second stage, we list their procedure steps in Algorithms 1 and 2, respectively. Moreover, we assume that the input data via was QAM or PSK mapped and then produced an OFDM signal x by way of the IFFT processor before entering the first stage.

In the first stage:

First, sort the OFDM signal x into $x^{\prime }$ order by each subcarrier amplitude, which is given by:(11)$$x^{\prime }=\left[x_{0}^{\prime },x_{1}^{\prime },\ldots ,x_{N-1}^{\prime }\right]$$
where $x_{0}^{\prime }$ represents the largest value in x, $x_{1}^{\prime }$ is the second largest value, $x_{N-1}^{\prime }$ is the minimum, and so on. Then, set the initial value of index *k* to zero.

In Step 3, set *n* to zero and the power allocation threshold β as $x_{k}^{\prime }$. In Step 4 and Step 5, use β to remedy the power of each subcarrier in *x*, then the PAPR of each revised corrected signal is calculated and stored in a sequence $q_{\mathrm{PAPR}}$. Next, in Step 6, re-give the β as $x_{k+1}^{\prime }$, and repeat the Steps from 4 to 5 until the PAPR of all different β is recomputed and stored. Finally, in Step 7, we select the power allocation threshold β with the lowest PAPR according to the criteria as
(12)$$\beta_{rough}=\beta_{\underset{0\leq k\leq N-1}{argmin}\left[q_{\mathrm{PAPR}}\left(k\right)\right]}$$

To obtain a reasonable proportion of PRT and reliable $\beta_{rough}$, we performed a Monte Carlo experiment on the algorithm in the first stage, and by way of the curve of the PAPR approach try to find the trustworthy $\beta_{rough}$ and relationship between the amount of PRT and subcarrier number. In the experiment, we adopt three different subcarrier lengths, which are 512, 1024, and 2048, respectively. Obviously, in Figure 8, we can find that no matter what the subcarrier number is, the amount of PRT is always about 8% of the subcarrier length, which is the best PAPR performance. Therefore, we know that a reasonable proportion of PRT estimation is 0.08, in the meantime, a reliable $\beta_{rough}$ can be produced in the first stage and the statistics expression can be denoted as follows:(13)$$\beta_{rough}=x_{n}^{\prime }{|}_{n\approx 0.08N}$$
**Algorithm 1** Algorithm for the first stageStep 1   Sort the IFFT output data X by tone magnitude from large to small, and denote              the sorted signal as $x^{\prime }=\left[x_{0}^{\prime },x_{1}^{\prime },\ldots ,x_{N-1}^{\prime }\right]$.Step 2   Set *k* to zero.Step 3   Set $\beta :=x_{k}^{\prime }$ and n:=0.Step 4   Detect each tone magnitude in the signal x, if $x_{n}\geq \beta$ and $x_{n}>0$ subtract β from              the tone, if $x_{n}\leq \beta$ and $x_{n}<0$, addition β to the tone. Otherwise, do not alter.Step 5   If n<N                 n:=n+1                 return Step 4              else                 Compute and store the PAPR of the revised signal into the sequence $q_{\mathrm{PAPR}}$.Step 6   If k<N                 k:=k+1              return Step 3Step 7   Select with minimum PAPR from the stored PAPR sequence, $q_{\mathrm{PAPR}}$, and then              produce relative $\beta_{rough}$.

In the second stage:

We know that since the $\beta_{rough}$ is a coarse power allocation threshold only, hence, in the second stage, we proposed an iteration scheme with a binary search procedure to trim the $\beta_{rough}$ to improve PAPR performance. Here, we use $\beta_{i}$ to represent the ith iteration of β and set $\beta_{1}$ to $\beta_{rough}$ for the initial iteration in the second stage. Next, we let the initial value of *n* to zero and $PAPR_{0}$ equal to $PAPR_{OFDM}$, where $PAPR_{OFDM}$ is the PAPR value of the original OFDM transmission. Thereupon, we can now deal with the amendment of power allocation, and the equation is as follows:(14)$$x_{n}=\left\{\begin{array}{ccc} x_{n}-\beta_{i} & , & x_{n}\geq \beta_{i}\\ x_{n}+\beta_{i} & , & x_{n}\leq -\beta_{i}\\ x_{n} & , & else \end{array}\right.;\;0\leq n\leq N-1$$
where *n* is the tone index, and *i* denote the numeric of iterations.

To avoid the random signal peak greater than multiple $\beta_{i}$, which results in converging difficulties, therefore, in Step 2, we utilize Equation (Equation 14) to limit the tone magnitude to less than $\beta_{i}$. Immediately after, to refine the power assignment and improve the BER performance of the receiver, in Step 3, we calculate PAPR and store the auxiliary signal, $a_{i}$, for each iteration. Therefore, we deposit the position of amendment power in the auxiliary signal and send it to the receiver simultaneously with the corrected signal to aid the receiver in retrieving the desired information sequence. Finally, in Step 4, we check whether the PAPR result calculated by Step 3 converges, if not, we the keep iteration using the binary search method.

Furthermore, in this paper, the definition of PAPR convergence is in a limited magnitude changes percentage within γ%, where γ is a small constant. Undoubtedly, the smaller γ is, the better the convergence, but the longer it takes to converge. For the convenience of explanation, we chose the constant γ% equal to 5% in our proposed algorithm.
**Algorithm 2** Algorithm for the second stageStep 1   Set $\beta_{1}:=\beta_{rough}$ and $PAPR_{0}:=PAPR_{OFDM}$Step 2   For n=0:N−1                 While $\left(|x_{n}\left|\geq \right|\beta_{i}|\right)$                      $x_{n}:=x_{n}\pm \beta_{i}$Step 3   Calculating the PAPR and storing the auxiliary signal $a_{i}$.Step 4   If $\frac{|PAPR_{i}-PAPR_{i-1}|}{PAPR_{i-1}}\times 100\%<\gamma \%$                 produce the output signal and the auxiliary signal $a_{i}$.              else                 set $\beta_{i+1}:=\beta_{i}/2$, and i:=i+1 then go to Step 2.

In short, for the overall algorithm, the rough of β can be simply taken from the *M*th largest amplitude of the subcarrier in the original transmission OFDM signal, and *M* can easily be obtained from the expression of 0.08N. It is worth mentioning that the rough of β was used for the initial of β for iteration in the second stage, possessing the advantage of simplifying the iteration number and then reducing the computational complexity and PAPR.

## 4. Simulation and Discussion

To clearly understand the implementation of each scheme, we give the parameter values of the experimental scenario in Table 2. It is also used as the specification parameter of the complexity of the numerical calculation example, simultaneously.

### 4.1. Experimental Results

Some simulations are performed to verify the PAPR and BER performance of the proposed I-TR scheme. The simulation scenario is an OFDM-based system with 16 and 64 quadrature amplitude modulation (QAM), and the subcarrier length *N* is equal to 512 and 1024, individually. Furthermore, we compare seven schemes including original OFDM, PTS [21], interleave-PTS [23], ABC-PTS [22], SLM [18], TI [27], TR [28], and the proposed scheme by comparing the performance of PAPR, for 16-QAM and 64-QAM is shown in Figure 9a,b and Figure 10a,b, where (a) and (b) are shown as *N* = 512 and *N* = 1024, respectively. To observe the approach of the convergence of the I-TR signal more clearly, the CCDF estimation and PAPR values are compared with the distinct number of iterations of I-TR, as shown in Figure 11 and Figure 12, respectively. Moreover, Figure 13 shows the power distribution of the original OFDM signal, and Figure 14a,b show the power distribution of first iteration and the convergent I-TR signal, respectively, under the complex plane.

First, from Figure 9a, we can find that the original OFDM has been seriously plagued by a higher PAPR, which also has the worst performance. As for the PTS-based family, such as the PTS [21], interleave-PTS [23], and ABC-PTS [22] algorithms, among them, the PAPR performance of ABC-PTS is reduced by about 3.31 dB compared to the original OFDM when CCDF is equal to 0.001. Although the PAPR performance of PTS-based schemes is similar, they are different in computational complexity. For the SLM method, the performance of SLM is slightly better than the above PTS-based scheme by only about 0.67 dB for the same environment, but the cost is prohibitive due to the high complexity. Moreover, the PAPR performance of the TI [27] algorithm is slightly better than SLM 0.45 dB, but it may be difficult for practical applications because of its high complexity. Now, considering both PAPR and complexity factors, the TR technology [28] is relatively feasible, because its PAPR performance is better than SLM 0.50 dB, which means it is better than the original OFDM 4.48 dB simultaneously, and the complexity is the lowest compared to the aforementioned schemes. Thus, we choose the TR technology as the base and then propose the I-TR scheme.

Obviously, for our proposed I-TR scheme, the first stage roughly provides the power allocation threshold for the second stage to approach the optimal power assignment, therefore, the PAPR performance is better than the TR technology [28] by 1.70 dB and better than the original OFDM by 6.18 dB, which has dramatically improved PAPR performance. Simultaneously, the I-TR scheme possesses the lowest complexity among the aforementioned schemes, which we will discuss in the following subsection.

Second, in Figure 9b, the number of subcarriers is increased to 1024. We can observe that the performance of each scheme almost maintains the same trend as Figure 9a, and the proposed I-TR scheme is still better than the original OFDM 6.44dB. Moreover, in Figure 10, the order of QAM is ascended to 64, and our proposed scheme still maintains the best among all the schemes. In the future, due to the high requirements for communication speed, signals with larger subcarrier numbers and higher modulation order will frequently appear in advanced wireless communication systems, and our proposed scheme still has some advantages.

Third, to analyze the PAPR performance of the iterative process of the I-TR scheme, we simulate the performance for a different number of iterations. It can be seen from Figure 11 that the CCDF estimation of the I-TR scheme is successively updated and indented to convergence in each iteration process. Obviously, it can reach convergence after about seven iterations, that is, the variation of PAPR refining is less than 5%, and we show the PAPR performance vs. the iterations number in Figure 12. Similarly, it can be found that it also approaches 4.5 dB PAPR after seven iterations, which echoes the verification in Figure 11 and also proves that I-TR is a convergent scheme.

To compare the power distribution interval between the original OFDM signal transmission and our scheme, we demonstrate them in Figure 13 and Figure 14, respectively. Wherein, Figure 14 shows the subcarrier power distribution of our proposed scheme, including the first iteration in (a), and that is has reached convergence in (b). Furthermore, the real part thresholds contours retracted from $\pm 1.74\times 10^{-1}$ to $\pm 2.72\times 10^{-3}$, and the imaginary part thresholds contours retracted from $\pm 1.82\times 10^{-1}$ to $\pm 2.85\times 10^{-3}$. Relatively, the PAPR is also reduced from 6.13 dB to 4.54 dB (see Figure 12). Hereupon, we can be inferred that the gap between the peak power and the average power is significantly reduced, and then effectively improves the performance of the PAPR. In contrast to the I-TR signal, for the original OFDM in Figure 13, because there is no threshold assistance, it hence brings about that the power distribution locus is disordered and not concentrated, undoubtedly, resulting in a higher PAPR.

Lastly, in the second stage of our proposed scheme, we conserve the position of amendment power in the auxiliary signal with side information to aid the receiver in retrieving the desired information sequence. To verify the BER performance, we show the simulation results in Figure 15. Thus, we can observe that our proposed I-TR scheme is better than the OFDM scheme $3.20\times 10^{-3}$ in BER when SNR is equal to 20 dB. Otherwise, when maintaining BER performance at $10^{-3}$, the SNR needs to be enhanced from 21 dB to 23 dB. From this, it can be seen the channel robustness of I-TR is better than OFDM. It is worth mentioning that we do not need to sacrifice the BER performance of the receiver to improve the PAPR at the transmitter.

### 4.2. Computational Complexity Analyze and Performance Discussion

#### 4.2.1. Complexity Expression Analyze

In this subsection, we evaluate the computational complexity of the proposed scheme and others in terms of the number of complex multiplications and additions (CMAs) required. To be more succinct, we have shown the parameter values in Table 2. Within that, *N* is the number of subcarriers, $G_{PTS}$, $G_{ABCPTS}$, and $G_{SLM}$ are the number of rotation factor combinations in the corresponding scheme, furthermore, *P* is the number of groups in the PTS algorithm, *u* is the number of additional equivalent constellation points in TI technology, *L* is the number of candidates peak countervail signals in the TR method, *M* is the estimated amount of PRT derived from the first stage, and *i* is the number of iterations in the second stage of I-TR. Furthermore, the mathematical functions of complex multiplication and addition (CMAs) required for each method are derived and described in the subsequent paragraphs.

First, we note that the SLM scheme uses point-to-point multiplication to multiply the number of subcarriers, *N*, by a set of $G_{SLM}$ rotation factor vectors, and performs an *N*-point IFFT for each sequence, requiring a total of multiplications and additions $G_{SLM}\left(N^{2}+N\right)$ and $G_{SLM}\left(N^{2}-N\right)$. To select the signal output with the best PAPR performance, we calculate and compare the PAPR of each sequence, $G_{SLM}\left(3N+3\right)$ and $\left[G_{SLM}\left(2\left(N-1\right)\right)+\left(G_{SLM}-1\right)\right]$ are additionally required, and after simplification, the total $G_{SLM}\left(N^{2}+4N+3\right)$ and $\left[G_{SLM}\left(N^{2}+N-1\right)-1\right]$ CMAs.

In contrast, the PTS scheme, the *N* subcarriers are equally divided into *P* clusters, and zero-padding is performed on the original length, *N*. After operations N-point IFFT, each cluster is multiplied by a constant rotation factor and the *P* clusters are added to obtain a PTS signal. We repeat the above steps to perform $G_{PTS}$ times, approximately required $\left[G_{PTS}\left(N^{2}/P+N\right)P\right]$ and $G_{PTS}\left[\left(N/P-1\right)NP+N\left(P-1\right)\right]$, calculate and compare the PAPR of the PTS signal need additionally, $G_{PTS}\left(3N+3\right)$ and $2G_{PTS}\left(N-1\right)$ CMAs and a total of $G_{PTS}\left[N^{2}+\left(P+3\right)N+3\right]$ and $\left[G_{PTS}\left(N^{2}+N-1\right)-1\right]$ CMAs. Furthermore, in the interleaved partition PTS algorithm in ref. [22], called interleave-PTS, the number of rotation factor groups is equivalent to 1/16 of all candidate groups, so the complexity is reduced to 1/16 of traditional PTS. In addition, the ABC-PTS technique doubles the number of clusters and uses $G_{ABCPTS}$ to represent the number of rotation factor combinations searched by the ABC algorithm, which after simplification requires $G_{ABCPTS}\left[N^{2}+\left(2P+3\right)N+3\right]$ and $\left[G_{ABCPTS}\left(N^{2}+N-1\right)-1\right]$ CMAs for multiplication and addition, respectively.

As for the TI scheme, it adds *u* equivalent points for each constellation point, and whenever the constellation point changes, an *N*-point IFFT operation is performed. Therefore, the requirement is approximately $\left[uN^{3}+3uN^{2}+4uN-1\right]$ and $\left[uN^{3}+uN^{2}-2\right]$ CMAs. For the TR scheme, the predesigned *L* groups of peak countervail signals are combined with the original signal in turn, and after *N*-point IFFT and PAPR operations are performed, then the optimal PAPR output is selected at last. Their multiplication and addition require approximately $L\left(N^{2}+3N+3\right)$ and $\left[L\left(N^{2}+N-1\right)-1\right]$ CMAs, respectively.

For our proposed I-TR scheme, first, we let integer constant m=⌊0.08N⌋ where ⌊⌋ is the floor function and find the sorted vector, $x^{\prime }$, which is ordered by tone magnitude. According to $x^{\prime }$ and the estimated amount of PRT, *m* to obtain $\beta_{rough}$. The required complexity can be expressed as an arithmetic progression formula $\left(N-1\right)+\left(N-2\right)+\ldots +\left(N-m\right)$. Let *i* denote the number of iterations before PAPR converges in the power allocation process. The total need for multiplications and additions are $\left[N^{2}+\left(3i+3\right)N+5i+3\right]$ and $\left[N^{2}+\left(4i+1\right)N-2\right]$, respectively. For easy reference, Table 3 shows the analytic expressions for the total numbers of CMAs required by the aforementioned schemes.

#### 4.2.2. Numerical Analysis of Computational Complexity

In this subsection, we compare the numerical analysis of computational complexity between the proposed scheme and the aforementioned schemes. We substitute the parameter values provided in Table 2 into the expressions in Table 3 and obtain the CMAs of each scheme when the subcarrier *N* is equal to 512 and 1024. Besides that, the numerical values are listed in Table 4, and a bar graph is shown in Figure 16.

From Table 4 and Figure 16, we can discover that the complexity of SLM and PTS-base schemes has an inseparable relationship with the number of rotation factor combinations. In general, more combination diversity can get better PAPR performance, but the cost will be reflected in the computational complexity. Thus it is a big problem to balance performance and complexity. In contrast, the TI is an example of sacrificing complexity to improve PAPR performance. Its PAPR performance is comparable to the TR technology, as observed in Figure 10, but the computational complexity of the TI scheme is $O(N^{3})$, which means that the TI is more suitable for the low subcarrier environment. Similar to the PTS-base family, the TR technique is also closely related to the number of candidate peak cancellation signals *L*. However, the TR scheme does not require a lot of candidate peak cancellation signals, *L*, to obtain a good PAPR. Therefore, the complexity can be effectively limited.

Fortunately, for our proposed scheme, the iterative process does not need to perform IFFT every time, which results in excessive complexity waste, and only a few iterations are required to achieve convergence. Therefore, although we also face the tradeoff between PAPR performance and complexity problems, we can improve this problem at the lowest cost compared to other methods. Certainly, the PAPR performance and complexity of I-TR are still the best among the aforementioned techniques.

To discuss each scheme in further detail, we first simultaneously consider Figure 9a and the case of N = 512 in Table 2. The SLM algorithm has as many as 538,448,895 CMAs. The second scheme, TI, has 269,486,081 CMAs, and the computational complexity of the TI scheme is not only lower than that of the SLM algorithm but it also outperforms SLM in PAPR performance, which shows that TI has more advantages in the environment of lower subcarriers. However, the TR scheme, which has nearly equal PAPR performance with TI, requires only 15,759,390 CMAs, which is 5.8% of the TI scheme. Additionally, the PAPR performance of the three methods in the PTS-base family is similar, and the interleave-PTS is the best in terms of complexity, using only 8,437,776 CMAs. Overall, the optimal solution seems to be the TR scheme. Although the complexity of interleave-PTS is lower than the TR scheme, the PAPR performance given by TR is more satisfactory. Finally, in terms of complexity and PAPR performance, our proposed I-TR method far exceeds the TR scheme.

Next, consider Figure 9b and the case of N = 1024 in Table 2. Unlike the above, the complexity of the TI scheme soars to 2,151,682,049 CMAs due to the increase of subcarriers, even slightly the 2,150,635,519 CMAs of the SLM algorithm. However, in the high subcarrier environment, neither the SLM nor the TI scheme are the optimal solution. The more competitive technologies are still 33,652,752 CMAs of interleave-PTS and 62,976,030 CMAs of the TR scheme. Similarly, our proposed I-TR scheme not only has better PAPR performance than the TR scheme, but it also has only 2,127,380 CMAs in complexity, which is about 97% lower than that of the TR scheme.

At first glance, the expressions in Table 3, except for the TI method, is ordered by $O(N^{3})$, and other schemes are ordered by $O(N^{2})$, but it is not the case, because $G_{PTS}$, $G_{ABCPTS}$, and $G_{SLM}$ parameters also have a great impact on the complexity. Therefore, only I-TR is really ordered by $O(N^{2})$, which can be clearly observed from the numerical value in Table 4 and Figure 16, especially for a larger number of subcarriers, *N*.

## 5. Conclusions

In this paper, for OFDM-based systems, an efficient side information-assisted two-stage I-TR scheme is proposed to improve the performance of PAPR, computational complexity without sacrificing BER. This study first uses a Monte Carlo method to yield precise initial estimates of the reasonable amount of PRT, *M* and rough power allocation threshold, $\beta_{rough}$, at low cost, and then iterates to refine the power assignment gradually so as to shrink the gap between the peak power and the average power to gain the near-optimum PAPR performance. In other words, with the developed ingenious two-stage I-TR processing scheme, the troublesome peak power can be succinctly suppressed, and simultaneously, the average power is not dropped severely to render precise subcarrier power allocation. Computer simulations show that the new two-stage I-TR transmitter, with modestly low complexity, provides superior performance compared with previous works in various scenarios.

Finally, since this research has the advantages of low complexity and low PAPR without sacrificing BER performance, it can provide a good choice for promoting PAPR performance and reducing the cost of transmitters in future wireless communication systems.

## Figures and Tables

**Figure 1 sensors-23-00950-f001:**
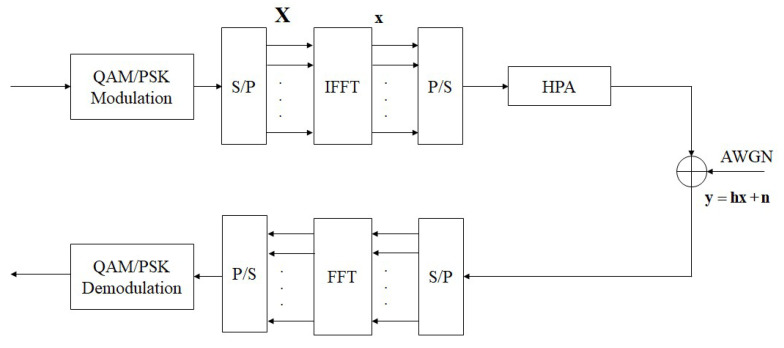
The block diagram of the fundamental of the OFDM system.

**Figure 2 sensors-23-00950-f002:**
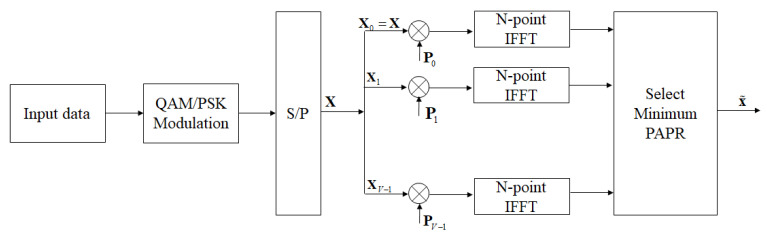
The block diagram of the SLM scheme in OFDM systems.

**Figure 3 sensors-23-00950-f003:**
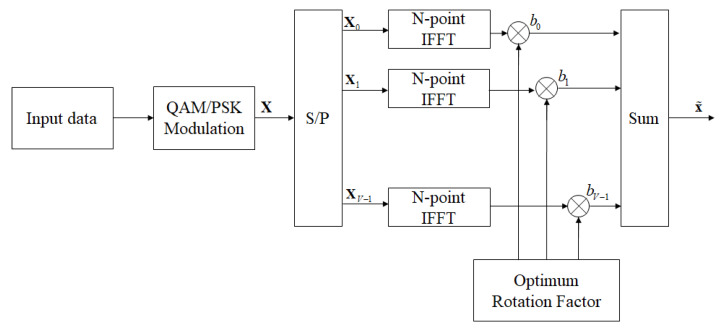
The block diagram of the PTS scheme in OFDM systems.

**Figure 4 sensors-23-00950-f004:**
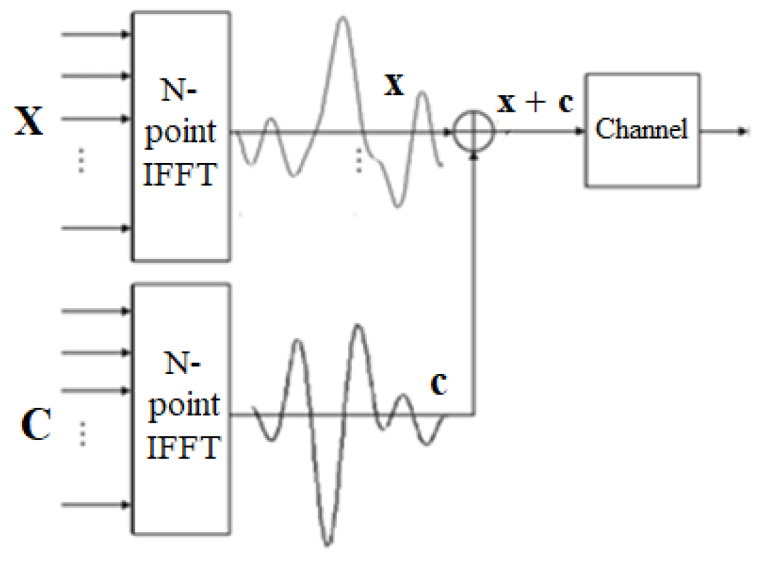
The block diagram of the TI scheme in OFDM systems.

**Figure 5 sensors-23-00950-f005:**
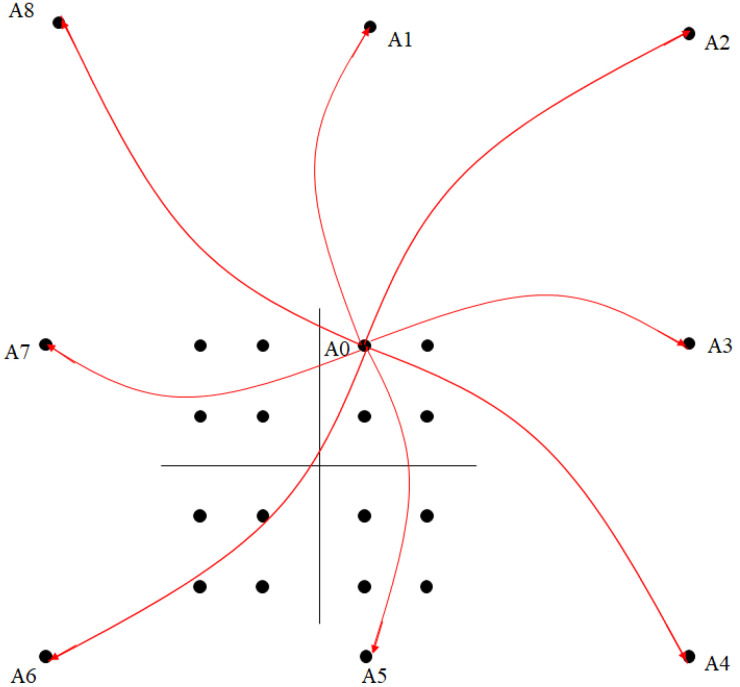
Expanded 16-QAM to 9 × 16-QAM constellation for tone injection (TI) scheme.

**Figure 6 sensors-23-00950-f006:**
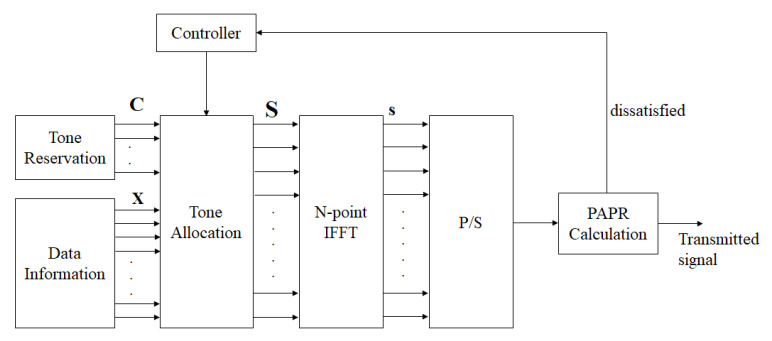
The block diagram of the TR scheme in OFDM systems.

**Figure 7 sensors-23-00950-f007:**
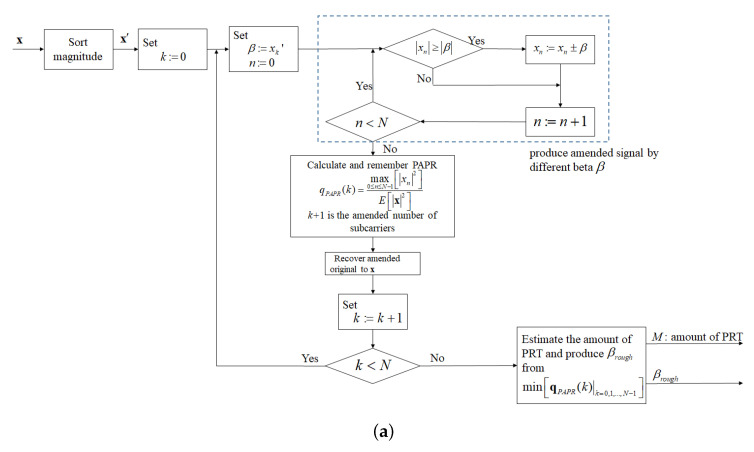
The block diagram of the proposed scheme in (**a**) the first stage and (**b**) the second stage.

**Figure 8 sensors-23-00950-f008:**
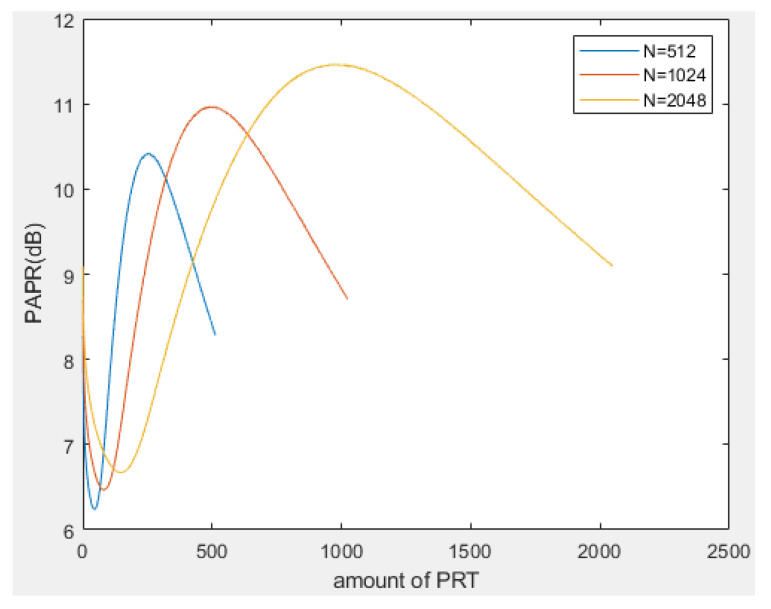
PAPR performance vs. amount of PRT for *N* = 512, 1024, 2048.

**Figure 9 sensors-23-00950-f009:**
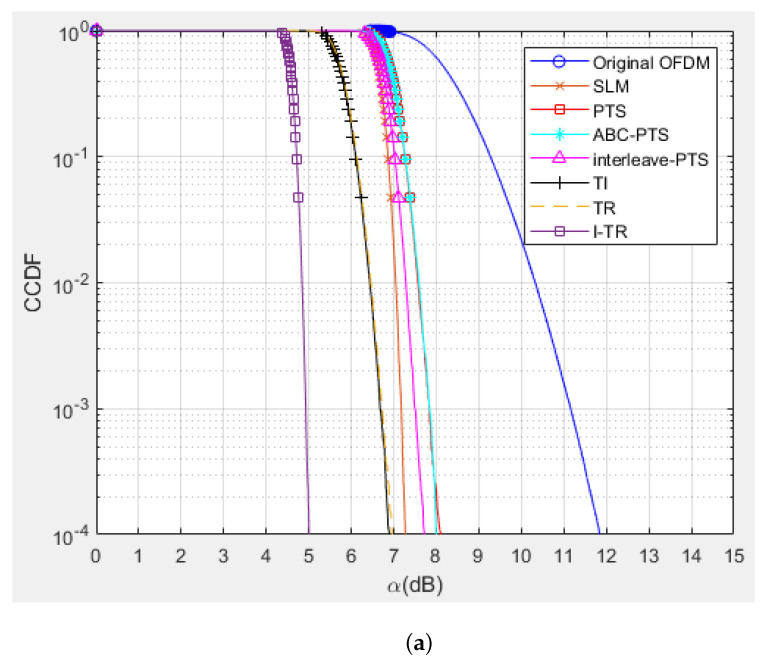
The PAPR performance of SLM [18], PTS [21], ABC−PTS [22], interleave−PTS [23], TI [27], TR [28], and I−TR for 16−QAM (**a**) *N* = 512 and (**b**) *N* = 1024.

**Figure 10 sensors-23-00950-f010:**
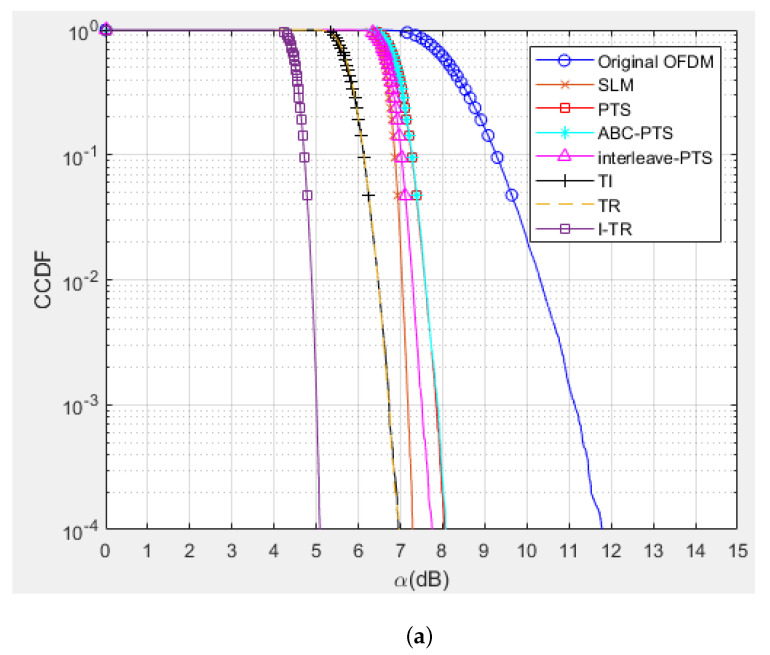
The PAPR performance of the different schemes for 64−QAM (**a**) *N* = 512 and (**b**) *N* = 1024.

**Figure 11 sensors-23-00950-f011:**
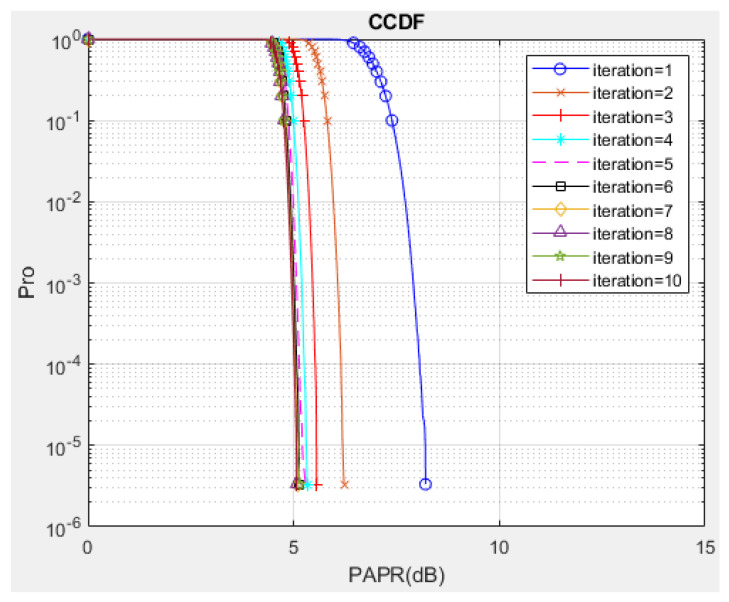
CCDF vs. α for a different iteration number for the proposed I−TR scheme.

**Figure 12 sensors-23-00950-f012:**
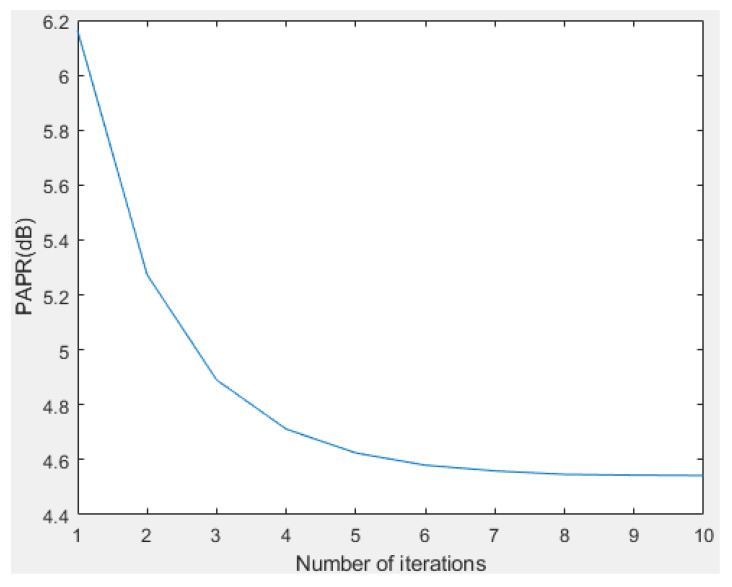
PAPR vs. the number of iterations for *N* = 1024.

**Figure 13 sensors-23-00950-f013:**
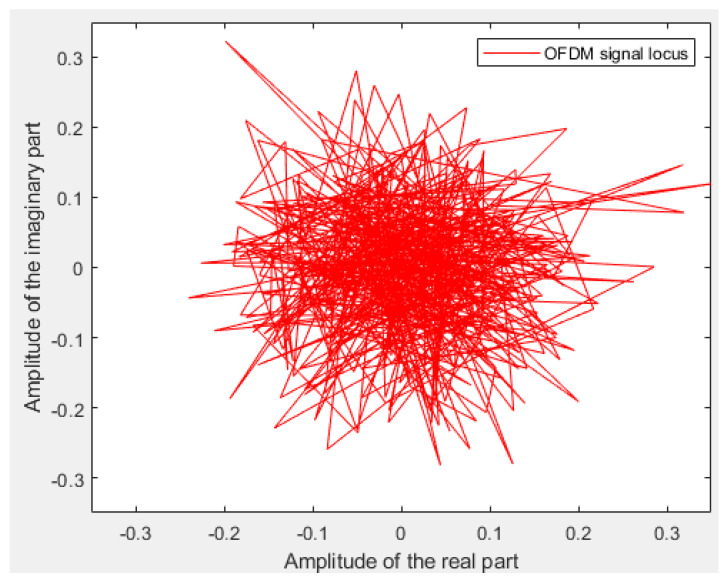
The power distribution locus for each subcarrier of the OFDM symbol.

**Figure 14 sensors-23-00950-f014:**
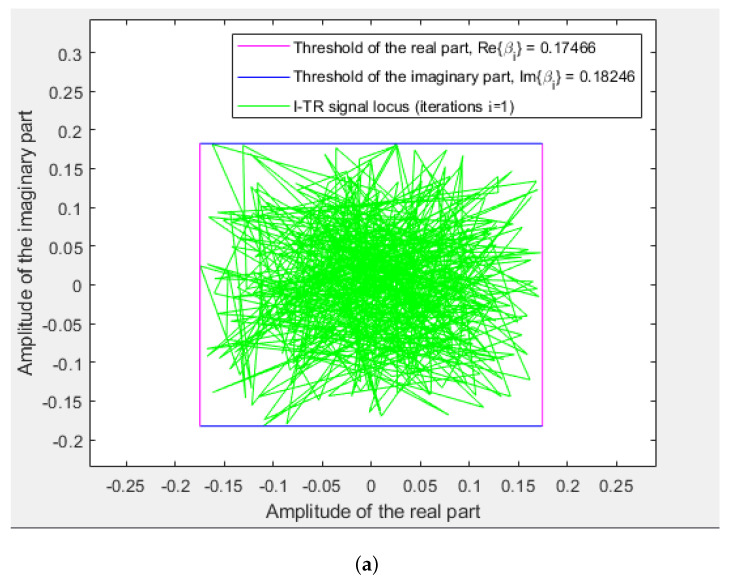
The power distribution locus for each subcarrier of the I−TR signal. (**a**) *i* = 1 (**b**) *i* = 7.

**Figure 15 sensors-23-00950-f015:**
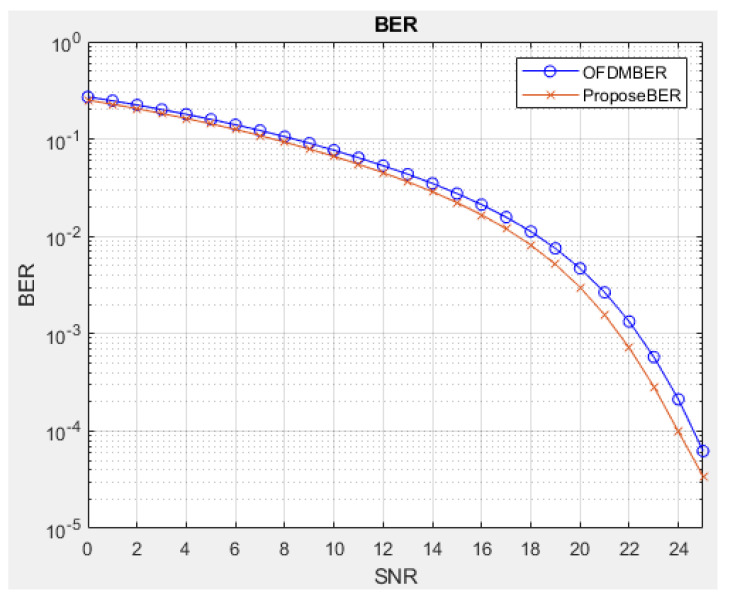
BER vs. SNR with OFDM and proposed scheme.

**Figure 16 sensors-23-00950-f016:**
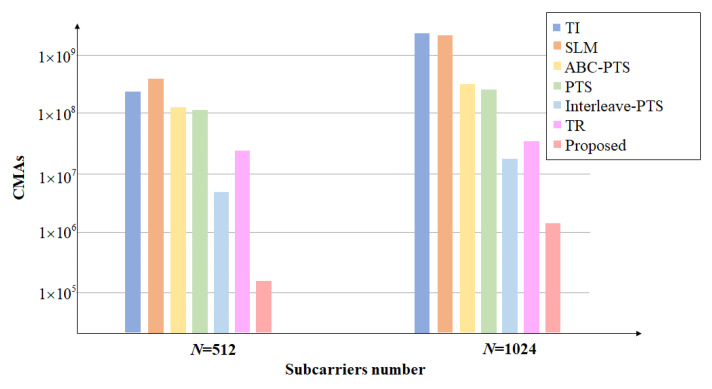
Computational complexity of SLM [18], PTS [21], ABC−PTS [22], interleave−PTS [23], TI [27], TR [28], and I−TR with *N* = 512 and *N* = 1024.

**Table 1 sensors-23-00950-t001:** The definitions of important notices and parameters.

Parameter Name	Description and Definition
β	Generalized power allocation threshold.
$\beta_{i}$	The power allocation threshold at the *i*th iteration in the second stage.
$\beta_{rough}$	The power allocation threshold produced in the first (initial) stage, can be used for the initial iteration in the second stage, in other words, which is equal to $\beta_{1}$.
α	The parameter of certain PAPR thresholds for the CCDF estimation.
$x^{\prime }$	The sorted signal of the original OFDM signal by each subcarrier magnitude is from large to small.
γ	The setting of the PAPR convergence range for a limited magnitude change percentage in the second stage.
$PAPR_{i}$	The PAPR at the ith iteration in the second stage.
$PAPR_{OFDM}$	The PAPR when the pure OFDM signal was transmitted, namely, is equal to $PAPR_{0}$.
$q_{\mathrm{PAPR}}$	The PAPR sequence for each different amount of PRT.
$a_{i}$	An auxiliary signal for remembering the amended magnitude and position at the ith iteration in the second stage.

**Table 2 sensors-23-00950-t002:** Simulation scenario and related parameters.

Parameter Name	Value
The number of experiment for Montecarlo.	100,000
The order of QAM.	16, 64
*N*, number of subcarrier of OFDM signal.	512, 1024
$G_{PTS}$, rotation factor group number of PTS scheme.	256
$G_{ABCPTS}$, rotation factor group number of ABC-PTS scheme.	256
$G_{SLM}$, rotation factor group number of SLM scheme.	1024
*P*, clusters number of PTS-based scheme.	4
*u*, additional equivalent constellation points number of TI method.	1
*L*, number of candidate peak countervail signals in TR scheme.	30
*i*, number of iterations in I-TR scheme.	7
*M*, estimated amount of PRT.	⌊0.08N⌋

**Table 3 sensors-23-00950-t003:** Comparison of the computational complexity of the aforementioned schemes.

Technology	Number of Complex Multiplications	Number of Complex Addition
SLM [18]	$G_{SLM}\left(N^{2}+4N+3\right)$	$\left[G_{SLM}\left(N^{2}+N-1\right)-1\right]$
Interleave-PTS [23]	$G_{PTS}\left[N^{2}+\left(P+3\right)N+3\right]/16$	$\left[G_{PTS}\left(N^{2}+N-1\right)-1\right]/16$
PTS [21]	$G_{PTS}\left[N^{2}+\left(P+3\right)N+3\right]$	$\left[G_{PTS}\left(N^{2}+N-1\right)-1\right]$
ABC-PTS [22]	$G_{ABCPTS}\left[N^{2}+\left(2P+3\right)N+3\right]$	$\left[G_{ABCPTS}\left(N^{2}+N-1\right)-1\right]$
TI [27]	$\left[uN^{3}+3uN^{2}+4uN-1\right]$	$\left[uN^{3}+uN^{2}-2\right]$
TR [28]	$L\left(N^{2}+3N+3\right)$	$\left[L\left(N^{2}+N-1\right)-1\right]$
Proposed	$\left[N^{2}+\left(3i+3\right)N+5i+3\right]$	$\left[N^{2}+\left(4i+1\right)N-2\right]$

**Table 4 sensors-23-00950-t004:** The numerical comparison of complexity for different schemes with *N* = 512 and 1024.

Scheme	CMAs *N* = 512	CMAs *N* = 1024
SLM [18]	538,448,895	2,150,635,519
ABC-PTS [22]	135,528,704	539,492,608
PTS [21]	135,004,416	538,444,032
Interleave-PTS [23]	8,437,776	33,652,752
TI [27]	269,486,081	2,151,682,049
TR [28]	15,759,390	62,976,030
Proposed	539,156	2,127,380

## Data Availability

Not applicable.

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
