# Peer review of "A Novel Low Complexity Two-Stage Tone Reservation Scheme for PAPR Reduction in OFDM Systems"

_sensors, 2023, doi:10.3390/s23020950_

Round 1

Reviewer 1 Report

This paper proposes a PAPR reduction scheme for OFDMA systems. Unfortunately, this paper is poorly prepared and I cannot recommend it for publication. Below are my concerns:

1. The results of PAPR evaluation and algorithm complexity are not convincing. The authors should not evaluate them separately, but show them together, so that the trade-off between the two can be well explained.

2. The authors need to explain the existing methods in more detail. It is recommended to add a related work section.

3. The numbers in Table 5 are not very informative. For complexity performance evaluation, the authors should express complexity as a function of N or other factors and evaluate it.

4. It is very difficult to understand how the schemes in Table 5 differ so much in complexity. The authors should explain with more specific examples how these numbers are obtained. Although the idea of ​​the proposed scheme is not so new and it is close to a simple power allocation scheme, the PARR performance is far better than the others. Where are these gains coming from? It is not well elaborated in this paper.

5. [MINOR] The current version is not well prepared. There are too many broken English sentences, and the structure is also not that good. For example, the introduction section is written in a single paragraph, and there are too many similar figures taking up too much space for no purpose. The authors need to make efforts for better representation.

Reviewer 2 Report

a) I recommend placing a paragraph at the end of the introduction where the contributions of the article are clearly written. For example:

The contributions of this article are:

1)

2)

The way they are currently established is unclear and confusing.

b) I recommend placing a table in the results and discussion section that clarifies the technical parameters used to obtain the results.

Reviewer 3 Report

The research presented in the article can be interesting to the radio communication system designers.

However, the manuscript requires much work in order to make it more clear and readable. Specifically:

The writing style should be significantly improved. There are many too long sentences, which are often unclear. The whole introduction is in a single, overly long paragraph.  Moreover, there are many grammatical and linguistic errors.

In the "Proposed scheme" section, the meaning of important notions and parameters should be clearly defined, for example \beta, \beta_{rough}, a_i and so on. The Figs. 3 a) and 3 b) do not precisely define the algorithms. Some elements should be clarified, e.g. there is a "Record PAPR" block in Fig. 3 a), but it is not shown, where the recorded PAPR is used then? The last, important block is "Select optimize \beta_{rough}", but how it is selected (it has to be guessed, basing on context and not so clear description in the text)? Then, some of the doubts are resolved by the descriptions in Table 1 and Table 2, however it is not obvious, why the same algorithm is presented in two different ways (Figure and then pseudo-code in the Table)? Also, there are some mistakes in the Tables too, e.g. in Step 4 in the Table 1, there is a condition: (xn<0 and xn>0), which is never true.

Due to all the doubts, the general idea is hard to grasp and it is questionable. 

In the results part, the method of presenting the power distribution by the signal trajectories in following iterations in a series of figures (Figs. 9 (a)...(g)) is not a good idea. All the Figures take much space, while presenting nothing quite important, just illustrating the PAPR decreasing with iterations (which is already shown in Fig. 8).

The concluding section should also be rewritten and expanded to more clearly indicate the research conclusions.

Round 2

Reviewer 1 Report

Much improvements have been made in this version, compared to the original one.